# Mechanisms of Embryonic Stem Cell Pluripotency Maintenance and Their Application in Livestock and Poultry Breeding

**DOI:** 10.3390/ani14121742

**Published:** 2024-06-09

**Authors:** Ziyu Wang, Wei Gong, Zeling Yao, Kai Jin, Yingjie Niu, Bichun Li, Qisheng Zuo

**Affiliations:** 1Joint International Research Laboratory of Agriculture and Agri-Product Safety of Ministry of Education of China, Yangzhou University, Yangzhou 225009, China; 211902331@stu.yzu.edu.cn (Z.W.); mx120220894@stu.yzu.edu.cn (W.G.); 211603122@stu.yzu.edu.cn (Z.Y.); 007838@yzu.edu.cn (K.J.); niuyj@yzu.edu.cn (Y.N.); yubcli@yzu.edu.cn (B.L.); 2Key Laboratory of Animal Breeding Reproduction and Molecular Design for Jiangsu Province, College of Animal Science and Technology, Yangzhou University, Yangzhou 225009, China

**Keywords:** ESCs, iPSCs, transcription factors, signaling pathways, epigenetic modifications

## Abstract

**Simple Summary:**

Embryonic stem cells have unique pluripotency and the potential to differentiate into any cell type in the body. This characteristic gives them great potential for application in the field of animal husbandry. Scientists can better control the differentiation process of embryonic stem cells by studying the maintenance mechanism of their pluripotency. In animal breeding, the use of embryonic stem cells can achieve the protection of animal species resources and genetic improvement of important traits, helping to cultivate animal breeds with faster birth growth and better meat quality. This not only improves agricultural production efficiency but also reduces the dependence on animals, which is more in line with ethical requirements and human production needs.

**Abstract:**

Embryonic stem cells (ESCs) are remarkably undifferentiated cells that originate from the inner cell mass of the blastocyst. They possess the ability to self-renew and differentiate into multiple cell types, making them invaluable in diverse applications such as disease modeling and the creation of transgenic animals. In recent years, as agricultural practices have evolved from traditional to biological breeding, it has become clear that pluripotent stem cells (PSCs), either ESCs or induced pluripotent stem cells (iPSCs), are optimal for continually screening suitable cellular materials. However, the technologies for long-term in vitro culture or establishment of cell lines for PSCs in livestock are still immature, and research progress is uneven, which poses challenges for the application of PSCs in various fields. The establishment of a robust in vitro system for these cells is critically dependent on understanding their pluripotency maintenance mechanisms. It is believed that the combined effects of pluripotent transcription factors, pivotal signaling pathways, and epigenetic regulation contribute to maintaining their pluripotent state, forming a comprehensive regulatory network. This article will delve into the primary mechanisms underlying the maintenance of pluripotency in PSCs and elaborate on the applications of PSCs in the field of livestock.

## 1. Introduction

Since the initial isolation of normal diploid pluripotent stem cells (PSCs) from mouse blastocysts in 1981 [1,2], later referred to as embryonic stem cells (ESCs), their long-term in vitro culture and applications have garnered significant attention [3]. The pluripotent nature of ESCs, allowing them to differentiate into nearly all cell types derived from the three germ layers, including germ cells, with the potential to return to their original niche in vivo [2,4], has positioned ESCs as an ideal cellular source for developmental biology or disease modeling, transgenic animal generation, and biotechnology applications. Despite their nearly limitless self-renewal capacity in vivo, this feature is challenging to maintain in vitro (prone to differentiation), requiring supplementation with exogenous factors (growth factors, small molecules) [5,6]. Nevertheless, the difficulty in establishing and maintaining ESC lines from livestock has hindered their widespread application. It was not until Shinya Yamanaka’s groundbreaking research that a new avenue for ESC application emerged [7]. Building upon the mechanisms of pluripotency maintenance, Yamanaka successfully reprogrammed mouse fibroblasts into ESC-like cells using Oct4, Sox2, Klf4, and c-Myc (*OSKM*) four factors, termed induced pluripotent stem cells (iPSCs) [7]. Induced pluripotent stem cells possess developmental potential comparable to ESCs, and as iPSC technology bypasses the use of embryos and can generate patient-specific pluripotent cells, it holds promise across various fields, particularly in medicine [6,7,8,9]. Although iPSC technology is currently more mature, the main research is centered around rodents and humans. In contrast, research in livestock and poultry is still in its infancy and faces more challenges [7]. Key challenges include restricted cell proliferation, dependence on exogenous factor expression, poor differentiation capacity, low reprogramming efficiency, and limited in vivo developmental potential [10]. Given that the generation of iPSCs is based on ESC pluripotency, understanding the mechanisms underlying the maintenance of embryonic stem cell pluripotency is crucial.

## 2. Mechanism of Pluripotency Maintenance of ESCs

### 2.1. Transcription Factor Related to Pluripotency

The maintenance of ESC pluripotency hinges on a regulatory network centered around transcription factors [11]. Key among these are OCT4, SOX2, and NANOG (*OSN*), which synergistically regulate genes essential for self-renewal and differentiation, thereby sustaining ESC pluripotency [12]. Research indicates that OCT4 is crucial for controlling ESC pluripotency, as it inhibits genes promoting differentiation (such as *hCG* and *IFN* genes) and activates those that support pluripotency. The correct expression level of OCT4 is vital, as ESCs retain their pluripotent state only when OCT4 is maintained at a normal level [13]. SOX2 is crucial for maintaining pluripotency, ranking just behind OCT4 [14]. Together, SOX2 and OCT4 synergistically sustain the pluripotency of ESCs by structurally interacting through their DNA-binding domains [15,16,17]. NANOG, a key transcription factor, regulates several downstream targets, including *Trp53*, which acts as a negative regulator of pluripotency [18]. *Trp53* inhibits the in vitro differentiation of ESCs and helps maintain their pluripotency [18]. Furthermore, the OCT4-SOX2 complex, along with secondary transcription factors like FOXD3, binds to the proximal promoter of NANOG to control its high expression levels [19,20]. The synergy among *OSN* and other transcription factors—including members of the KIF family, C-MYC, TFCP2L1, TFE3, YAP, ID1/2/3, DAX1, ESRRB, TBX3, and PRDM14 [21]—is essential for maintaining the pluripotent state of ESCs. These factors not only regulate *OSN* expression but also impact pluripotency through their specific roles in extracellular signaling pathways such as leukemia inhibitory factor (LIF), bone morphogenetic protein 4 (BMP4), and Wnt. Leukemia inhibitory factor, for instance, enhances the expression of *Nanog* and *Oct4*, which are critical for stem cell maintenance [22,23] (Figure 1).

### 2.2. Signal Pathway

Indeed, signaling pathways play a crucial role in maintaining stem cell pluripotency in ESCs. They primarily contribute, either directly or indirectly, to the regulation of *OSN* expression. Leukemia inhibitory factor is indispensable for long-term culture of ESCs from different species in vitro and maintains the pluripotency of ESCs mainly through the enhancement of NANOG and OCT4 expression [22,24]. The significance of the LIF signaling pathway in the maintenance of cellular pluripotency stems from its broad interactions with pathways that promote pluripotency, such as Janus kinase-signal transduction and transcription activation 3 (JAK-STAT3), phosphoinositide 3-kinase (PI3K)-AKT, and YES-yes-associated protein (YAP), as well as those that encourage cellular differentiation, including mitogen-activated protein kinase–extracellular signal-regulated kinases (MAPK-ERK) [25,26]. When the JAK-STAT3 signaling pathway is activated by LIF signaling, it triggers the phosphorylated Janus kinase (PJAK)-STAT3-BCL3 cascade, which prevents cell differentiation while enhancing the expression of OCT4 and NANOG [27]. Concurrently, other related transcription factors, such as KLF4, TFCP2L1, ESRRB, and SALL4, are activated to support the core regulatory network essential for maintaining pluripotency [28,29]. *Tfcp2l1*, a target gene of STAT3, plays a pivotal role in transmitting LIF signaling to the key transcription factors necessary for maintaining pluripotency [30]. The PI3K-AKT signaling pathway can enhance pluripotency and cell proliferation by inhibiting MAPK-ERK signaling [31,32], which otherwise promotes endodermal differentiation, and by boosting the expression of TBX3, NANOG, and C-MYC [24]. The YES-YAP, MAPK-ERK, BMP, and Wnt signaling pathways also contribute to maintaining cellular pluripotency by regulating the expression of *OSNs* through direct or indirect mechanisms. These signaling pathways are not entirely independent in their role in maintaining pluripotency. For instance, *Dusp9*, a downstream target of BMP signaling, interacts with mitogen-activated protein kinase (MAPK) by deactivating extracellular signal-regulated kinases (ERKs) [33]. Meanwhile, Wnt signaling can either positively or negatively influence the expression of OCT4 and NANOG through β-catenin and TCF3, respectively. However, β-catenin serves as a negative regulator of TCF3 [34,35,36]. Notably, the regulation of β-catenin by Wnt signaling depends on GSK3β, which can, in turn, be targeted by PI3K-AKT signaling to regulate Wnt signaling expression [37,38] (Figure 1).

### 2.3. Epigenetic Modification

Epigenetic factors are essential in the synergy between signaling pathways and pluripotent transcription factors in regulating the maintenance of pluripotency, such as DNA methylation, histone modification [39], and ATP-dependent chromatin remodeling [40,41]. Typically, high levels of DNA methylation inhibit gene expression, while low levels facilitate it [42]. Moreover, the *OSN* transcription factors, which are influenced by epigenetic modifications, also have the ability to regulate changes in epigenetic modifications. For example, *Oct4* is not only a major target of NANOG-TET1 regulation but also influences the expression of *Tet1* [43]. Research has discovered a dual role of *OSNs* in the regulation of pluripotency maintenance. On the one hand, they can act as transcription factors involved in DNA methylation and histone modifications, such as histone methylation and acetylation [44]. That is to say, *OSN* transcription factors can maintain the high expression level of genes involved in the maintenance of pluripotency and inhibit the expression of genes related to the promotion of cellular differentiation by altering the epigenetic modification status of the target. On the other hand, the expression of *OSNs* could be regulated by epigenetic modifications [45,46]. Another compelling piece of evidence demonstrating the ability of *OSN* to influence genome-wide epigenetic modifications involves Xist RNA, a cis-regulator essential for X chromosome inactivation (XCI) [47]. This process functions by suppressing gene expression and facilitating the deposition of the inactive chromatin marker H3K27me3 [47]. It has been observed that transcription factors such as OCT4, SOX2, and NANOG can bind to the coding region of Xist RNA, consequently inhibiting Xist transcription and impacting the status of XCI [47]. Furthermore, epigenetic modification serves as a critical link in the signaling pathway that regulates the expression of pluripotency transcription factors. Downstream transcription factors in the signaling pathway control the expression of pluripotency genes by recruiting enzymes for epigenetic modification. For example, transcription factors such as STAT3, SMAD1, and TCF3 target the *OSN* and serve as downstream effectors for the LIF, BMP4, and Wnt signaling pathways [17,48]. SMAD1, by interacting with epigenetic modifiers like the histone acetyltransferase p300/CREB-binding protein (P300/CBP), activates target gene expression [49]. This acetylation facilitates the development of an open chromatin structure, enhancing the expression of pluripotency genes such as *Nanog* and *Oct4* [50]. Under LIF signaling, STAT3 recruits TET1 and JMJD2 to the promoter regions of pluripotency genes, promoting demethylation. This process not only increases open epigenetic marks on DNA and histones but also boosts the expression of pluripotency genes, thereby sustaining the pluripotency of ESCs [27,51]. During Wnt signaling, β-catenin interacts with t-cell factor/lymphoid enhancer-binding factor (LEF/TCF) family factors to form a complex that attracts epigenetic modification enzymes, including HDAC1 [51]. This complex converts from a transcriptional repressor to an activator, boosting the expression of pluripotency factors like NANOG and OCT4 [52]. Additionally, other transcription factors from various signaling pathways also regulate the expression of pluripotency genes through direct or indirect epigenetic modifications. These include PRDM14, esBAF, and BRG [12,53,54]. Overall, maintaining stem cell pluripotency is a complex and intricately controlled process that involves multiple signaling pathways and epigenetic factors (Figure 1).

**Figure 1 animals-14-01742-f001:**
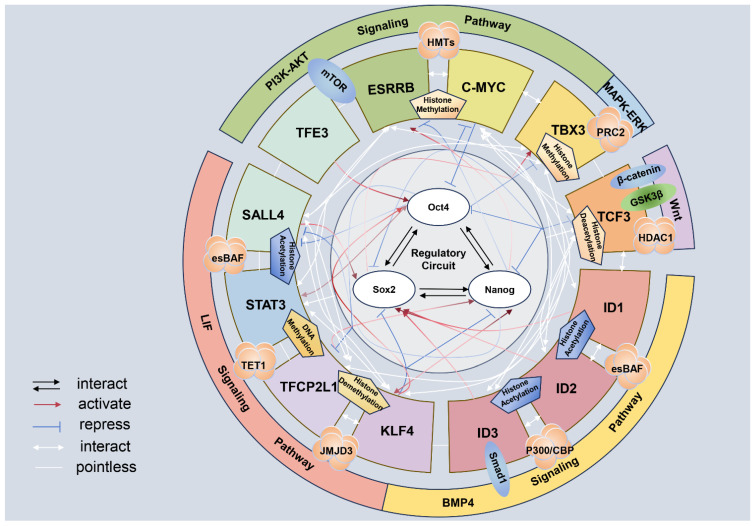
Regulatory networks of epigenetics, signaling pathways, and transcription factors in the maintenance of pluripotency in embryonic stem cells. The leukemia inhibitory factor (LIF) signaling pathway can recruit TET1, esBAF complex, and JMJD2, which, through the methylation of DNA, the demethylation, and acetylation of histones, act on downstream transcription factors such as STAT3, TFCP2L1, SALL4 and KLF4 and promote the expression of the core pluripotency genes *Oct4/Sox2/Nanog* (*OSN*) [48,51]. The bone morphogenetic protein 4 (BMP4) signaling pathway, through the activation of the primary signal transduction molecule SMAD1, participates in transcriptional regulation. SMAD1, in conjunction with acetyltransferases p300/ CREB-binding protein (P300/CBP) and the esBAF complex [12,55], activates downstream transcription factors ID1, 2, 3 through acetylation, thereby activating the expression of core transcription factors like SOX2 [33]. When the Wnt signaling pathway activation acts on β-catenin and GSK3β, then β-catenin translocates to the nucleus, forms a complex with transcription factors of the t-cell factor/lymphoid enhancer-binding factor (TCF/LEF) family, such as TCF3, and recruits epigenetic modifying enzymes like HDAC1 [51,52]. Through the deacetylation of histones, it regulates the expression of core pluripotency genes *OSN* [33]. The mitogen-activated protein kinase–extracellular signal-regulated kinases (MAPK-ERK) signaling pathway, by regulating transcription factors like TBX3, recruits polycomb repressive complex 2 (PRC2) to mediate histone methylation, thus maintaining the repressed state of pluripotency genes through inhibitory regulation. The phosphoinositide 3-kinase (PI3K)-AKT signaling pathway, by mediating downstream transcription factors TFE3, ESRRB, C-MYC, and recruiting histone methyltransferases (HMTs), regulates *OSN* through histone methylation [33,56]. In addition to the signaling pathway, there are also regulatory roles between these downstream transcription factors. According to the STRING database analysis, there is an interplay between TFCP2L1, ESRRB, SALL4, TBX3, and KLF4, while the latter four interact with STAT3; STAT3 can also affect TCF3. At the same time, C-MYC interacts with ID1, 2, and 3, as well as TCF3. TCF3 can also be affected by ID1, 2, and 3. Additionally, the downstream transcription factors of these pathways can also interact with the core transcription factors *OSN*. STAT3 and OCT4 mutually activate each other; TFCP2L1 can activate NANOG while being repressed by OCT4, and KLF4 can repress OCT4 and NANOG and also activate OCT4 and NANOG. It can also be activated by NANOG. ID1, 2, and 3 can activate SOX2; TCF3 mutually inhibits OCT4 and NANOG and represses SOX2; TBX3 is inhibited by OCT4 but activated by SOX2; C-MYC is repressed by SOX2 and also represses the regulation of OCT4; NANOG can both activate and repress ESRRB; TFE3 promotes the expression of OCT4; and SALL4 is repressed by OCT4 and NANOG and activates SOX2 [56,57,58,59]. There are also mutual regulatory relationships between the *OSN* genes themselves [56]. In the figure, the red line represents activation, and the direction of the arrow indicates the direction of activation; the blue line represents repression, and the direction of the “T” indicates the direction of repression; the black arrowheads and white arrowheads indicate interactions between transcription factors; and the white straight lines indicate modification connections, with no special meaning.

## 3. Application of Pluripotent Maintenance Mechanism (Induced Pluripotent Stem Cells)

Takahashi and Yamanaka successfully reprogrammed fibroblasts into iPSCs, thereby pioneering iPSC research [7,60]. Induced pluripotent stem cells exhibit similar pluripotency to ESCs and offer significant advantages in terms of cost, ease of generation, and practical application [61,62]. They represent a direct application of research into the mechanisms of long-term pluripotency, effectively replacing ESCs in research contexts and showing potential for significant impact in livestock breeding and other fields. The initial concept behind iPSC technology involved inducing somatic cells to revert to a pluripotent state through the introduction of exogenous pluripotency factors (OSKM) [63], endowing them with ESC-like properties. With continuous improvements in the iPSC induction system, the efficiency of in vitro induction has progressively increased. This advancement underscores the critical reliance on the developmental history of research into pluripotency maintenance mechanisms.

Research on ESCs and iPSCs has advanced significantly since 2006 when Shinya Yamanaka’s team first reported their groundbreaking results in the journal *Cell* [7,60]. Their research demonstrated the reprogramming of mouse fibroblasts into iPSCs using four transcription factors cloned into viral vectors [60,62,64,65]. Since then, various methods for generating iPSCs have been explored and refined, including different combinations of transcription factors, small molecule compounds, and non-integrating viral vectors. These advancements have not only increased the efficiency of iPSC generation but also reduced potential safety risks [63,66,67,68,69,70,71,72,73,74,75]. Like ESCs, induced pluripotent stem cells are pluripotent, capable of self-renewing and differentiating into any cell type across the three germ layers. The role of transcription factors in generating iPSCs is critical. Initially, Yamanaka’s team used OCT4, SOX2, C-MYC, and KLF4 [63]. Subsequent research indicated that C-MYC and KLF4 could be replaced with other transcription factors like NANOG and LIN28 [66]. Moreover, small-molecule compounds such as valproic acid and GSK3β inhibitors have been employed to enhance iPSC generation [70]. Currently, methods for iPSC generation primarily encompass retroviral, lentiviral, adenoviral, plasmid transfection, transposon, and recombinant protein approaches [76]. Among these, retroviral and lentiviral vectors stand out as the most [77] commonly employed techniques. However, these methods carry the risk of integrating into the host genome. As a result, researchers are exploring non-integrative approaches, including the use of small molecules and mRNA for cellular reprogramming [73,78]. Epigenetic changes are crucial in the iPSC reprogramming process. This includes alterations in DNA methylation, histone modifications, and chromatin remodeling [40,41], which collectively shift the gene expression pattern of adult cells towards a pluripotent state.

## 4. Application of Pluripotent Stem Cells

The livestock and poultry industry is pivotal to global food security and the agricultural economy. As the world’s population grows and consumption patterns evolve, there is an increased demand for more efficient and sustainable livestock and poultry production systems. Gene editing tools have become instrumental in this sector, leading to the development of livestock and poultry with enhanced disease resistance, such as pigs resistant to porcine reproductive and respiratory syndrome virus (PRRSV), cows resistant to Mycobacterium bovis, and chickens resistant to avian leukemia virus (ALV) [79]. Furthermore, stem cell technologies, particularly ESCs and iPSCs, offer promising new avenues for the conservation of livestock and poultry genetic resources, improvement of critical traits, and disease management, leveraging their capabilities in cell differentiation and regenerative potential.

### 4.1. Reproduction: Theoretical Basis for the Application of Pluripotent Stem Cells

As the understanding of pluripotency in PSCs deepens, their contributions to reproductive biology and regenerative medicine continue to expand, particularly in the area of creating functional germ cells from PSCs. These advancements not only enrich reproductive medicine theory but also offer novel strategies for hastening genetic improvement in livestock and conserving endangered species. In 2023, researchers reported a significant breakthrough in the *Cell Stem Cell* journal, detailing the efficient and stable generation of ESCs from bovine blastocysts [80]. This study highlighted that these ESCs can not only proliferate indefinitely but also differentiate into functional sperm and eggs in vitro. Additionally, porcine iPSCs have been successfully differentiated into primordial germ cell-like cells (PGCLCs) [81,82]. These PGCLCs were co-cultured with early testis cells and successfully produced functional sperm capable of fertilizing eggs to produce offspring [83]. These discoveries open up new possibilities for genetic improvement and the conservation of domestic animals. Well-established systems for forming PGCLCs from PSCs are now in place for species like pigs, horses, rabbits, and mice [84,85,86]. These PGCLCs express crucial germ cell markers such as SOX17, BLIMP1, and TFAP2C [84,87], mirroring the biological characteristics and functions of in vivo-sourced primordial germ cells (PGCs). This advancement offers a new approach to gamete production in the laboratory using PSC technology, facilitating the use of these gametes for in vitro fertilization and embryo transfer [87,88]. This method could significantly speed up the genetic improvement of livestock. Moreover, this technique shows promise for conserving endangered species by enabling the production of healthy offspring in controlled environments without necessitating direct contact with adult animals [89]. Despite the potential, these technologies are still in their nascent stages, and further research is necessary to overcome technical challenges, such as enhancing efficiency, ensuring gamete quality and health, and addressing ethical and legal concerns. Nonetheless, these developments mark an exciting frontier with the potential to revolutionize our understanding and applications of reproduction and genetics in the future (Figure 2).

### 4.2. Conservation and Restoration of Rare Species (or Individuals)

In recent years, the rapid advancement of production has led to a troubling trend of endangerment and extinction of various species, with the situation becoming increasingly dire. Pluripotent stem cells, including ESCs and iPSCs, are vital for the conservation of endangered animal resources [90,91]. While traditional conservation methods like seed banking and the cryopreservation of sperm and eggs have been successful for plants and some animal species, they have limitations in fostering reproduction and genetic diversity among endangered animals [92,93]. This is particularly true for species that have become extinct or are on the brink of extinction in the wild, where acquiring sufficient reproductive cells for standard assisted reproductive technologies (ARTs) is exceedingly difficult. The development of ESC and iPSC technologies offers a potential solution to this issue [94]. Induced pluripotent stem cell technology is especially promising in endangered species conservation because it avoids the use of embryos, which can be impractical or controversial in many situations [95,96]. For example, in 2021, researchers successfully generated iPSCs from the skin cells of the endangered northern white rhinoceros using this technology [97]. Following this, Hayashi M. et al. [98] developed a culture system that used iPSCs from both the northern and southern white rhinoceros to create PGCLCs, providing a method to produce functional gametes from northern white rhinoceros iPSCs and potentially preventing their extinction. This achievement not only highlights the potential of iPSC technology in the conservation of endangered species but also sets a foundation for possible future “resurrection” projects. More recently, Masafumi Katayama and colleagues [99] applied seven reprogramming factors (M3O, SOX2, KLF4, C-MYC, NANOG, LIN28, and KLF2) to establish iPSCs in primary fibroblasts from three endangered bird species: the Okinawan rail (*Hypotaenidia okinawae*), the Japanese rock ptarmigan (*Lagopus muta japonica*), and Blakiston’s fish owl (*Bubo blakistoni*), offering new perspectives for conservation biologists and avian stem cell specialists in their future endeavors. As cryobiology continues to advance, primary fibroblasts from domestic animals such as cattle, buffalo, sheep, goats, and pigs have been successfully cryopreserved [100]. Importantly, these cell bank-derived somatic cells retain the potential to be reprogrammed, enabling their use in endangered species conservation through iPSC technology when necessary (Figure 2).

### 4.3. Cell Agriculture (Cultured Meat, etc.)

Cellular agriculture aims to create animal-based products such as meat, eggs, leather, or fur in the laboratory without harming or killing live animals [101,102]. This method offers a more environmentally friendly alternative to traditional farming, with iPSC technology enabling the scalability of animal product creation [103]. Skin and fur derived from animal iPSCs are poised to replace animal-sourced leather and fur. Common types of leather from cattle or pigs, which are by-products of meat breeds, lack breed-specific traits, resulting in uneven quality. In contrast, synthetic animal leather (derived from iPSCs) provides consumers and industries a first step away from industrialized agriculture [88]. Pluripotent stem cells are considered ideal cell sources for producing cultured meat without sacrificing animals [104]. Bovine stem cells were used to create the world’s first lab-grown meat burger, served at a press conference in London in 2013 [105]. Mark Post and his team marked a significant advancement in cellular agriculture by demonstrating that lab-cultured meat is feasible [106]. The burger patty consisted of 10,000 individual muscle fibers, differentiated from cultured bovine muscle stem cells using tissue engineering techniques [103]. Recently, bovine umbilical cord blood cells were reprogrammed into iPSCs and subsequently differentiated into muscle and fat cells [102,107]. However, the cells still require animal products for proliferation, so the end product is not entirely “animal-free” [107]. Recent developments in cell culture methods have led to the creation of xenogeneic-free and feeder-free stem cell cultures, reducing or completely eliminating animal products in their protocols to meet future regulatory restrictions and improve quality control processes [108]. Cultured meat aims to produce green meat proteins efficiently, offering significant advantages over traditional meat in terms of improving agricultural resource utilization [109], reducing greenhouse gas emissions [110], enhancing animal welfare [105], and improving the nutritional content of the products [100]. The use of livestock iPSCs in cellular agriculture (cultured meat production) is proposed as a clean and prominent alternative to lessen the global burden of livestock farming [88,111] (Figure 2).

### 4.4. Cell Therapy and Disease Models

Pluripotent stem cell technology has been applied to numerous animal models to assess the efficacy of innovative cell therapies [112,113]. Large animal models are particularly advantageous in preclinical trials, as they more accurately simulate human physiological and pathological processes, evaluating the efficacy and risks of treatments [114,115]. Additionally, they help determine effective cell dosages and assess the integration of transplanted cells within host organs [116]. These models provide invaluable tools for studying human diseases and treatments in the laboratory [117,118,119].

Pigs, due to their physiological and genetic similarities to humans, are widely used as animal models for evaluating new cell therapies [120]. For instance, pig iPSCs (piPSCs) can differentiate into retinal cells in vitro and integrate into the pig’s body post-transplantation, serving as important models for studying retinal diseases [121,122]. Moreover, pigs are valuable models for researching other conditions such as diabetes and Alzheimer’s disease [121,123,124]. As a novel cell therapy tool, pig iPSCs have shown great potential in tissue engineering and regenerative medicine. The piPSC-derived osteoblast-like cells have been shown to improve the trabecular and cortical bone structure in tibial fractures [125]. Pig iPSCs have also been demonstrated to aid in myocardial regeneration, where undifferentiated piPSCs directly injected into the myocardium significantly reduced infarct size, decreased local perfusion issues, and increased angiogenesis [126].

Cattle are used as models for studying human female fertility [127]. Research has shown that bovine iPSCs (biPSCs) can differentiate into cells with a breast epithelial phenotype and have the same potential for tissue regeneration as breast stem cells [128], which provides new ideas for the treatment of breast tumors. Equine iPSCs (eiPSCs) have been differentiated into various cell and tissue types for disease modeling, including neurons [129], tendons [130], myotubes [131], and osteoblasts [132]. Functional neurons derived from eiPSCs have been produced, capable of generating action potentials in vitro through functional calcium channels [133]. Keratin-forming cells derived from eiPSCs (eiPSC-KCs) can produce skin grafts for wound treatment [134,135]. Canine iPSCs (ciPSCs) can differentiate into mesenchymal stem cells (MSCs) and subsequently into chondrocytes and osteoblasts, serving as effective models for studying canine osteoarthritis [136]. In laboratory research, scientists have successfully used goose iPSCs (giPSCs) to study energy and resistance strategies against Newcastle disease virus (NDV) infection [137]. Results indicated that giPSCs could be infected by NDV, and in some cases, their resistance to the virus even increased, despite these cells not possessing innate viral resistance. This suggests that they can be used as tools for studying the viral lifecycle and potential treatment methods. Additionally, goose iPSCs have been used to manufacture a special type of virus—a replication-deficient virus. This virus halts replication mid-process, so it does not cause disease but retains the capacity to stimulate the host immune system. Scientists have created a replication-deficient version of the H5N1 influenza virus using giPSCs [138], which can be used in vaccine research because it can provoke an immune response without causing severe illness. One potential advantage of using giPSCs for vaccine production is that they might be safer than traditional chicken egg or chicken embryo culture systems, as they do not support complete viral replication, thereby reducing the risk of viral escape [139]. Furthermore, giPSCs can be cultured on a large scale in the laboratory, offering the potential for mass production of vaccines (Figure 2).
Figure 2Different applications of embryonic stem cells in the field of animal husbandry. The applications of embryonic stem cells (ESCs) in the field of animal husbandry are primarily categorized into four distinct areas: reproduction and genetics, conservation of endangered species, cell agriculture, and cell therapy and disease modeling. In reproduction and genetics, embryonic stem cells and induced pluripotent stem cells are mainly used to generate functional germ cells such as sperm and eggs [80,81,82,83]. Embryonic stem cells and induced pluripotent stem cells are used for freezing and storing somatic cells other than germ cells to conserve endangered animals [90,91,92,93]. Embryonic stem cells and induced pluripotent stem cells can be used in cellular agriculture to produce synthetic products such as meat and eggs and to make food products [101,102]. Porcine-induced pluripotent stem cells can be used as a model for researching retinal diseases, and they also help in myocardial regeneration [121,122,126]. Bovine iPSCs (biPSCs) can differentiate into cells with a breast epithelial phenotype and have the same potential for tissue regeneration as breast stem cells [128], which provides new ideas for the treatment of breast tumors. Equine iPSCs (eiPSCs) can differentiate into a variety of cell and tissue types such as tendons, which are used for disease modeling and can be used to treat equine musculoskeletal injuries in vivo [140,141]. Canine iPSCs (ciPSCs) can serve as a useful model for studying canine osteoarthritis [136].
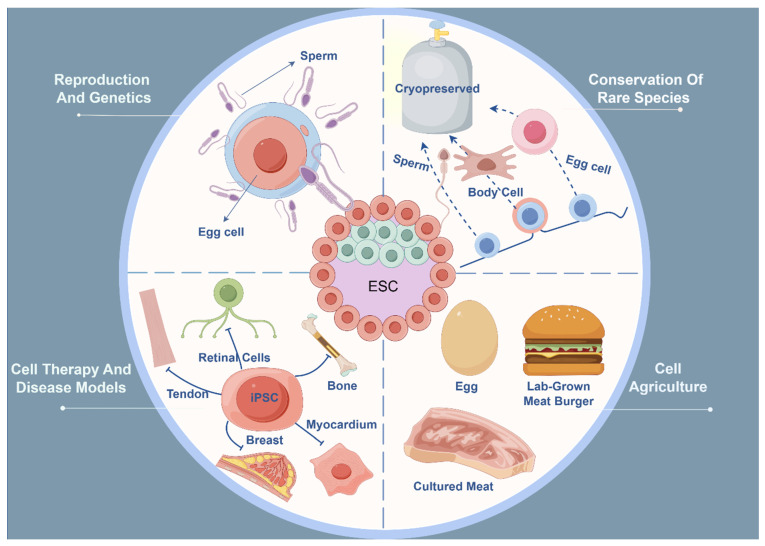


## 5. Conclusions and Perspectives

Embryonic stem cell research is steadily advancing, with varying degrees of progress in understanding the mechanisms that maintain pluripotency across different livestock species. Despite these differences, exploratory findings have laid a solid foundation for further in-depth research. The potential of ESCs is broad, with applications validated and utilized in regenerative medicine, disease prevention, and reproduction. These applications are not only of significant importance to the livestock industry, where they can improve the efficiency of animal production and breeding, enhance animal health, and strengthen the adaptability of livestock to environmental changes, but they also have a profound impact on modern medicine. Embryonic stem cells can provide materials for human pharmaceuticals, model human diseases, aid in the study of disease mechanisms and drug screening, and be used to repair or replace damaged tissues and organs. As research progresses, the scope of ESC applications is expected to expand further. We are committed to developing new uses for ESCs, especially with the help of gene editing technologies. Through precise genetic manipulation, not only can we cultivate livestock breeds with stronger resistance and superior economic traits, thereby greatly promoting the marketization and large-scale production of the livestock industry and enhancing its overall competitiveness and sustainable development capabilities, but we can also explore the use of these technologies to treat genetic diseases or create new types of organisms with specific characteristics, thereby reducing production costs and promoting research for health and safety. In the future, we will continue to strengthen basic research, expand application fields, promote the deep integration of ESCs with the livestock industry and modern medicine, and contribute to the development of modernization, greening, and intelligentization. At the same time, we will pay close attention to ethical and legal issues to ensure the sustainable development of ESC research and applications.

## Data Availability

All data used are included within the manuscript.

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
