# Peer review of "Mechanisms of Embryonic Stem Cell Pluripotency Maintenance and Their Application in Livestock and Poultry Breeding"

_animals, 2024, doi:10.3390/ani14121742_

Round 1

Reviewer 1 Report

Comments and Suggestions for Authors

In this review, the authors aim to discuss the mechanisms behind embryonic stem cell pluripotency and their potential applications in animal husbandry. The review is timely, informative, and relevant. However, I have several issues and suggestions that need to be addressed:

In the first sentence of the introduction, the authors reference the landmark work on the isolation of mouse pluripotent stem cells (PSCs) in 1981, but the original reference is not properly cited. The authors should ensure accurate citation of all references through the whole manuscript.

The sentences at lines 125-128 are confusing and need clarification to improve readability and comprehension.

 In Figure 1, the pink arc on the left and the green arc on the top appear to be displaced (too close to the inner circle) and can be corrected. Additionally, the arrows in the figure are not explained in the legend, particularly the white arrows between the neighboring genes in the second layer of the ring-shaped diagram. Please provide explanations for these elements in the legend. 

The illustration of Figure 2 is inconsistent with the figure legend. This discrepancy must be addressed.

It is interesting and relevant that the authors list their future perspectives and research plans in the "Conclusions and perspectives" section, particularly regarding the application of embryonic stem cell (ESC) studies in animal husbandry. However, it is suggested that the authors share their thoughts from a broader perspective to appeal to a wider audience.

Comments on the Quality of English Language

Moderate editing of language is required.

Author Response

Reviewer1-1 In the first sentence of the introduction, the authors reference the landmark work on the isolation of mouse pluripotent stem cells (PSCs) in 1981, but the original reference is not properly cited. The authors should ensure accurate citation of all references through the whole manuscript.

Thanks for your suggestion. We fully agree with the comments of the reviewer for not citing the original references correctly in manuscript. We have re-checked and corrected these errors in the whole manuscript.

Reviewer1-2 The sentences at lines 125-128 are confusing and need clarification to improve readability and comprehension.

Thanks for your suggestion. We have revised the sentences as below:

Research has discovered a dual role of OSNs in the regulation of pluripotency maintenance. On one hand, they can act as transcription factors involved in DNA methylation and histone modifications, such as histone methylation and acetylation. That is to say, OSN transcription factors can maintain the high expression level of genes involved in the maintenance of pluripotency and inhibit the expression of genes related to the promotion of cellular differentiation by altering the epigenetic modification status of the target. On the other hand, the expression of OSNs could be regulated by epigenetic modifications.

Reviewer1-3 In Figure 1, the pink arc on the left and the green arc on the top appear to be displaced (too close to the inner circle) and can be corrected. Additionally, the arrows in the figure are not explained in the legend, particularly the white arrows between the neighboring genes in the second layer of the ring-shaped diagram. Please provide explanations for these elements in the legend. 

Thanks for your suggestion. Many thanks to the reviewer for correctly pointing out the deficiencies in Figure 1. We have revised Figure 1 and provide more detail in the legend.

Reviewer 2 Report

Comments and Suggestions for Authors

Comments about the manuscript:

“Mechanisms of embryonic stem cell pluripotency maintenance and their application in livestock and poultry breeding”

The pluripotency of embryonic stem cells makes it possible to obtain all cell types, which leads to their use in the field of breeding, which is less limited from this point of view than human and rodent embryonic stem cells. To understand and control the differentiation process of these embryonic stem cells, it is necessary how pluripotency is maintained. In the review proposed here, the authors take stock of this question.

This well-written review is very well researched. It should be useful to researchers but also to teachers and students interested in animal biology and veterinary sciences. I will only make a few minor remarks.

Page 2, lines 53-54. “remains challenging when compared to rodents and humans”: I'm not sure human stem cells aren't a challenge!

Page 2, lines 65-66. “such as hCG and IFN genes": use italics to write gene names: check throughout the manuscript.

Page 4, figure1. This excellent figure is not called out in the text.

Page 7, lines 283-284. Use italics to write the names of genera and species: Hypotaenidia okinawae, Lagopus muta japonica, Bubo blakistoni.

Page 9, figure 2. This excellent figure is not called out in the text.

Author Response

Reviewer2-1 Page 2, lines 53-54. “remains challenging when compared to rodents and humans”: I'm not sure human stem cells aren't a challenge!

Thanks for your suggestion. There are also challenges associated with human induced pluripotent stem cells, but human embryonic stem cells and induced pluripotent stem cells are more intensively researched and what I want to emphasize here is that generating iPSCs from farm animals is more challenging compared to the booming development of research on rodents and human iPSCs.

To avoid misunderstanding, we have made the following changes:

Although iPSC technology is currently more mature, the main research is centred around rodents and humans. In contrast, research in livestock and poultry is still in its infancy and faces more challenges.

Reviewer2-2 Page 2, lines 65-66. “such as hCG and IFN genes": use italics to write gene names: check throughout the manuscript.

Thanks for your suggestion. The manuscript has been modified accordingly. Thank you for your valuable input!

Reviewer2-3 Page 4, figure1. This excellent figure is not called out in the text.

Thanks for your suggestion. We missed the labelling, thanks for the heads up!

Reviewer2-4 Page 7, lines 283-284. Use italics to write the names of genera and species: Hypotaenidia okinawaeLagopus muta japonicaBubo blakistoni.

Thanks for your suggestion. Corrections have been made to the original manuscript

Reviewer2-5 Page 9, figure 2. This excellent figure is not called out in the text.

Thanks for your suggestion. We have called out the figure2 in manuscript.

Round 2

Reviewer 1 Report

Comments and Suggestions for Authors

The manuscript has notably improved after the revision. However, there are still some issues that need to be addressed:

1.      Section Titles: Sections 3 and 4 have identical titles at lines 185 and 222. Please revise the titles to accurately reflect the content of each section.

2.      Figure 1: The modifications to Figure 1 have improved its clarity. However, there are still some issues with the connections and arrow lines:The color codes used in the figure are not annotated in neither the figure itself nor the legend. Please add annotations for the color codes to improve clarity. Several lines are bifurcated, making it difficult to follow the pathways. The authors should carefully redesign the figure to ensure it is both informative and concise.

3.      Figure 2:The reference should be added to the legend for proper citation.In the lower-left quadrant, the "T"-shaped arrowheads might not indicate inhibition, as is commonly understood. Please clarify this in the legend. All "Disease Models" reviewed are in preclinical research using livestock animals as model species. However, the legend includes inaccurate descriptions. For example, it states, "Bovine induced pluripotent stem cells can be differentiated into epithelial cells of mammary cells and applied to tumor patients undergoing mastectomy," which is not accurate. The cited reference (#127) only reports the generation of iPSCs from bovine epithelial cells and demonstrates their potential for generating bovine mammary tissue. Their wording and figure description overstate the results. If there are additional studies supporting these claims, please cite them. Otherwise, revise the statement to accurately reflect the findings. Additionally, the legend states, "Horse and dog induced pluripotent stem cells can be used for disease modeling and treatment of bone-like diseases." Similarly, the authors should cite related literature supporting the claim that "Horse and dog induced pluripotent stem cells can be used for the treatment of bone-like diseases." Furthermore, the term "Bone-like disease" is not a common term and needs further explanation or to be rephrased. The authors should apply this standard throughout the entire manuscript to improve scientific writing and ensure precision and readability.

Comments on the Quality of English Language

Minor editing of English language is required

Author Response

The manuscript has notably improved after the revision. However, there are still some issues that need to be addressed:

Reviewer1-1 Section Titles: Sections 3 and 4 have identical titles at lines 185 and 222. Please revise the titles to accurately reflect the content of each section.

Thank you for your suggestion. We apologise that we incorrectly repeated the headings in lines 185 and 222, we have corrected the error and the two correct headings are Application of pluripotent maintenance mechanism (induced pluripotent stem cells), Application of pluripotent stem cells in in livestock and poultry.

Reviewer1-2 Figure 1: The modifications to Figure 1 have improved its clarity. However, there are still some issues with the connections and arrow lines:The color codes used in the figure are not annotated in neither the figure itself nor the legend. Please add annotations for the color codes to improve clarity. Several lines are bifurcated, making it difficult to follow the pathways. The authors should carefully redesign the figure to ensure it is both informative and concise.

Thanks for your suggestion. We have annotated the colour codes in the figure and legend, and modified the diverging paths to make the figure clearer.

Figure 1. Regulatory networks of epigenetics, signaling pathways and transcription factors in the maintenance of pluripotency in embryonic stem cells. The LIF signaling pathway can recruit TET1, esBAF complex and JMJD2, which through the methylation of DNA, the demethylation and acetylation of histones, act on downstream transcription factors such as STAT3, TFCP2L1, SALL4 and KLF4, and promote the expression of the core pluripotency genes OSN (Oct4/Sox2/Nanog)[48, 51]. The BMP4 signaling pathway, through the activation of the primary signal transduction mol-ecule SMAD1, participates in transcriptional regulation. SMAD1, in conjunction with acetyl-transferases P300/CBP and the esBAF complex[12, 55], activates downstream transcription factors ID1, 2, 3 through acetylation, thereby activating the expression of core transcription factors like SOX2[33]. When the Wnt signaling pathway activation acts on β-catenin and GSK3β, then β-catenin translocates to the nucleus, forms a complex with transcription factors of the TCF/LEF family, such as TCF3, and recruits epigenetic modifying enzymes like HDAC1[51, 52]. Through the deacetylation of histones, it regulates the expression of core pluripotency genes OSN[33]. The MAPK-ERK signaling pathway, by regulating transcription factors like TBX3, recruits PRC2 to mediate histone methylation, thus maintaining the repressed state of pluripotency genes through inhibitory regulation. The PI3K-AKT signaling pathway, by mediating downstream transcription factors TFE3, ESRRB, C-MYC, and recruiting HMTs, regulates OSN through histone methyla-tion[33, 56]. In addition to the signalling pathway, there are also regulatory roles between these downstream transcription factors. According to the STRING database analysis, there is an inter-play between TFCP2L1, ESRRB, SALL4, TBX3 and KLF4, while the latter four interact with STAT3, STAT3 can also affect TCF3. At the same time, C-MYC interacts with ID1, 2 and 3, as well as TCF3. TCF3 can also be affected by ID1,2 and 3. Additionally, the downstream transcription factors of these pathways can also interact with the core transcription factors OSN. STAT3 and OCT4 mutually activate each other, TFCP2L1 can activate NANOG while being repressed by OCT4, KLF4 can repress OCT4 and NANOG, and also activate OCT4 and NANOG. It can also be activated by NANOG. ID1, 2, 3 can activate SOX2, TCF3 mutually inhibits OCT4 and NANOG and represses SOX2, TBX3 is inhibited by OCT4 but activated by SOX2, C-MYC is repressed by SOX2 and also represses the regulation of OCT4, NANOG can both activate and repress ESRRB, TFE3 promotes the expression of OCT4, and SALL4 is repressed by OCT4 and NANOG and activates SOX2[56-59]. There are also mutual regulatory relationships between the OSN genes themselves[56]. In the figure, the red line represents activation and the direction of the arrow in-dicates the direction of activation, the blue line represents repression and the direction of the “T” indicates the direction of repression, the black arrowheads and white arrowheads indicate inter-actions between transcription factors, and the white straight lines indicate modification connections, with no special meaning.

Reviewer1-3 Figure 2:The reference should be added to the legend for proper citation.In the lower-left quadrant, the "T"-shaped arrowheads might not indicate inhibition, as is commonly understood. Please clarify this in the legend. All "Disease Models" reviewed are in preclinical research using livestock animals as model species. However, the legend includes inaccurate descriptions. For example, it states, "Bovine induced pluripotent stem cells can be differentiated into epithelial cells of mammary cells and applied to tumor patients undergoing mastectomy," which is not accurate. The cited reference (#127) only reports the generation of iPSCs from bovine epithelial cells and demonstrates their potential for generating bovine mammary tissue. Their wording and figure description overstate the results. If there are additional studies supporting these claims, please cite them. Otherwise, revise the statement to accurately reflect the findings. Additionally, the legend states, "Horse and dog induced pluripotent stem cells can be used for disease modeling and treatment of bone-like diseases." Similarly, the authors should cite related literature supporting the claim that "Horse and dog induced pluripotent stem cells can be used for the treatment of bone-like diseases." Furthermore, the term "Bone-like disease" is not a common term and needs further explanation or to be rephrased. The authors should apply this standard throughout the entire manuscript to improve scientific writing and ensure precision and readability.

Thanks for your suggestion. As the commenter mentioned, the “T” arrows do not indicate inhibition, but rather an introductory function, and the graphic arrows that are quoted and cause misunderstandings have been explained in the figure notes. We apologise for the inaccuracies mentioned by the reviewer and have made the following changes.

bovine iPSCs (biPSCs) can differentiate into cells with a breast epithelial phenotype and have the same potential for tissue regeneration as breast stem cells[128], which provides new ideas for the treatment of breast tumours. Equine iPSCs (eiPSCs) can differentiate into a variety of cell and tissue types such as tendons, which are used for disease modelling and can be used to treat equine musculoskeletal injuries in vivo [140, 141]. Canine iPSCs (ciPSCs) can serve as a useful model for studying canine osteoarthritis[136].